



# Brief Communication: Dynamic magnification factors for tree blow-down by powder snow avalanche air blasts

Perry BARTELT[1], Peter BEBI[1], Thomas FEISTL[2], Othmar BUSER[2], and Andrin CAVIEZEL[1]

[1]WSL Institute for Snow and Avalanche Research SLF, Flüelastrasse 11, 7260 Davos Dorf, Switzerland
[2]Lawinenwarnzentrale im bayerischen Landesamt für Umwelt, Hessstrasse 128, 80797, Munich, Germany

*Correspondence to:* Perry Bartelt (bartelt@slf.ch)

**Abstract.** We study how short duration powder avalanche blasts can break and overturn tall trees. Tree blow-down is often used to back-calculate avalanche pressure and therefore constrain avalanche flow velocity and motion. We find that tall trees are susceptible to avalanche air blasts because the duration of the air blast is near to the period of vibration of tall trees, both in bending and root-plate overturning. Dynamic magnification factors for bending and overturning failures should therefore be
5   considered when back-calculating avalanche impact pressures.

## 1   Introduction

In this paper we develop a simple method to determine the dynamic response of trees to impulsive loads. This is an important problem in natural hazards engineering where historical evidence of forest destruction/tree breakage is often used to evaluate the potential avalanche hazard. Any indication of forest damage is particularly valuable to avalanche engineers because it helps
10   define the destructive reach of an extreme and infrequent event. Fallen tree stems delineate the spatial extent of an avalanche and create a natural vector field indicating the primary flow direction of the movement (Fig. 1). The age of the destroyed trees can be additionally used to link the historical observations to avalanche return period. In many cases observations of forest destruction are the only data the engineer has to quantify avalanche danger.

The problem with using evidence of tree destruction for avalanche mitigation planning is that a simple relationship between
15   avalanche impact pressure and tree failure is difficult to establish. Recent observations by Feistl et al. (2015) suggest that the magnitude of the avalanche impact pressure is strongly related to the avalanche flow regime. Although long recognized that dense flowing avalanches can easily break, overturn and uproot trees (Bartelt and Stöckli, 2001; Feistl et al., 2014), tree destruction by powder avalanche air blasts has received less attention. A mechanical understanding of how trees are blown-down by powder avalanche blasts would allow engineers to quantify powder avalanche pressures from case studies and
20   historical records.

Here we develop a mechanical model to predict the natural frequency of trees subject to full-height air-blasts of powder snow/ice avalanches. We assume two deformation modes: stem bending and root-plate overturning, see figs. 2.1 and 2.2. The ratio of the natural tree frequency to the frequency of the avalanche air-blast defines the dynamic magnification factor $D$ (Clough and Penzien, 1975). This value is used to magnify the non-impulsive loadings $D > 1$ to account for he increase in stress



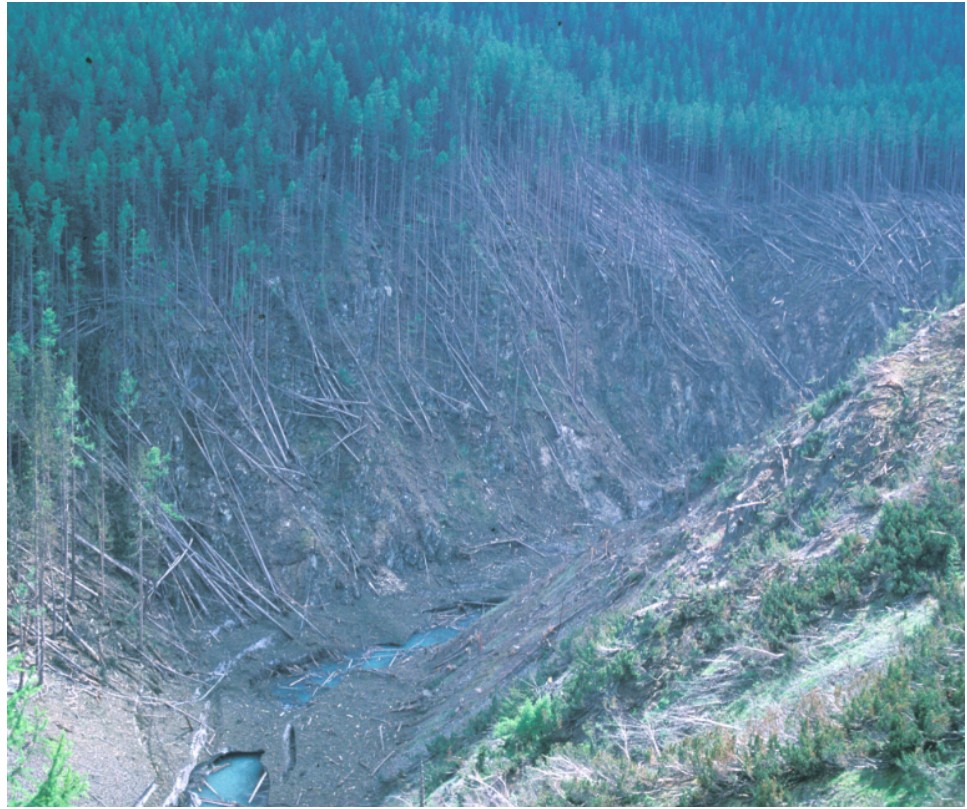

**Figure 1.** Tree (spruce) breakage caused by the air blast of a powder avalanche, Zernez, Switzerland, 1999. The trees failed in bending and root-plate overturning. Phoograph: Peter Bebi, SLF.

under an impulsive load. The eigenfrequency of the tree which is a function of the tree height, stiffness and mass distribution between the stem and branches and therefore depends on forest age and tree species. We show that dynamic magnification factors for fully grown trees are large indicating that mature forests are especially vulnerable to powder snow avalanches.

## 2  Tree Response to Impulsive Loading

5  Measurements on real avalanches reveal that the air-blast is intermittent and of short duration, lasting only a few seconds (Grigoryan and others, 1982; Sukhanov and Kholobayev, 1982; Sukhanov, 1982). When a powder avalanche hits a forest the ice-dust cloud is typically moving with velocities in excess of 50 m/s (similar to extreme wind gusts). The height of the cloud is equal, if not larger, than the height of the tree, i.e. $H > 20$ m. The pressure blast thus acts over the entire width and height of the tree, producing large bending moments in the stem and straining the root base plate. The impulsive character of the powder

10  avalanche air-blast, however, magnifies the static stress state (Clough and Penzien, 1975). The fallen tree stems often point in





the direction of the flow, indicating that the trees had little time to sway and react to blast and that the inertial effects are of considerable importance.

To calculate the dynamic magnification factor $D$ we first make three simplifying assumptions. Firstly, the air blast can be expressed as a sine wave impulse with duration time $t_0$. Moreover,

$$F(t) = F_0 \sin \overline{\omega} t, \tag{1}$$

where $\overline{\omega}$ is the circular frequency of the loading $\overline{\omega} = \pi / t_0$. The magnitude of the force $F_0$ is

$$F_0 = p_0 A = \frac{1}{2} c_d \rho U_{max}^2 A, \tag{2}$$

where $p_0$ is the amplitude of the avalanche pressure given by the density of the powder cloud $\rho$, the form drag coefficient of the tree $c_d$ and the maximum velocity of the blast $U_{max}$. The tree area over which the blast acts is denoted $A$, typically given by the tree height $H$ and effective tree width $W$. Thus, if the cloud density and velocity are known as well as the tree geometry, the magnitude of the applied blast force $F_0$ is given.

Secondly, after the loading time $t_0$ the tree vibrates freely with natural frequency $\omega$. The natural frequency is found using the Rayleigh quotient method (Clough and Penzien, 1975), which assumes the deflected form is known (but not the magnitude of deformation). The assumption of a deflected shape reduces the tree to a single degree of freedom system. The frequency is found by equating the maximum strain energy $V_{max}$ to the maximum kinetic energy $T_{max}$ developed during the tree response. By calculating the strain and kinetic energy produced by the avalanche blast, we find the generalized stiffness $K$ and generalized mass $M$ of the tree:

$$\omega^2 = \frac{K}{M}. \tag{3}$$

The natural frequency for two different deformation modes, stem bending $\omega_{sb}$ and root-overturning $\omega_{ro}$ will be determined in the next sections.

In both cases the total tree height is $H$. Tree mass is divided into two parts: the stem mass $m_s$ (a mass per unit length of the tree kg/m) and the total mass of the branches $M_b$ (kg). The branch mass, including the mass of needles, is lumped at the tree center-of-mass. The mass $M_b$ can include the mass of snow held by the branches and thus, like the tree elasticity, have some seasonal variation. As we assume a constant stem diameter $d$ the stem mass per unit length is

$$m_s = \rho_t A_t \tag{4}$$

with

$$A_t = \frac{\pi}{4} d^2. \tag{5}$$

The density of the stem wood is $\rho_t$. For both the bending and overturning cases, the concentrated load $F_0$ acts at the tree center-of-mass, which is located a distance $a$ from the ground (see figures 2.1 and 2.2).

Finally, the third assumption, the maximum response of the tree will be reached before the damping forces can absorb the energy of the air blast. Only the undamped response to a short duration blast is considered.





## 2.1 Eigenfrequency: Tree bending mode

For the case of tree bending the deformation $x(z)$ at height $z$ is given by (see Fig.2.1)

$$x_1(z) = X_0\psi_1(z) = \frac{Fa^2(3H-a)}{3EI}\left[\frac{3az^2 - z^3}{2a^2(3H-a)}\right] \quad \text{for} \ \ z \le a \tag{6}$$

and

$$x_2(z) = Z_0\psi_2(z) = \frac{Fa^2(3H-a)}{3EI}\left[\frac{3za^2 - a^3}{2a^2(3H-a)}\right] \quad \text{for} \ \ z > a \tag{7}$$

where $E$ is the modulus of elasticity of the tree stem and $I$ is the moment of inertia. These equations for the lateral tree deformation are found by assuming the tree is a statically determinate cantilever-type structure fixed at the base to the ground (see Fig. 2.1). The largest bending moment in the tree is found at the tree base, $z=0$. The quantity $X_0$ is the static deformation under the blast load $F$

$$X_0 = \frac{Fa^2(3H-a)}{3EI}. \tag{8}$$

The moment of inertia is taken for circular stem sections

$$I = \frac{\pi d^4}{64}. \tag{9}$$

The maximum potential strain energy in bending is

$$V_{max} = \frac{1}{2}X_0^2\int\limits_0^a EI(z)x_1^2(z)dz = \frac{1}{2}\frac{3EI}{a^2(3H-a)}X_0^2. \tag{10}$$

In the bending case the tree is firmly rooted in the ground and strain energy is stored in the tree stem between the ground and the point of load application $z = a$. The tree stem above $z > a$ is stress free, swaying back and forth as a rigid body. The maximum kinetic energy $T_{max}$ is composed of two parts containing the stem mass $T_{max}^s$ and the branch mass $T_{max}^b$ of the tree, $T_{max} = T_{max}^s + T_{max}^b$ :

$$T_{max}^s = \frac{\omega^2}{2}\int\limits_0^a m_s x_1^2(z)dz + \frac{\omega^2}{2}\int\limits_a^H m_s x_2^2(z)dz = \frac{1}{280}m_s\frac{\left[105H^3 - 105aH^2 + 35Ha^2 - 2a^3\right]}{(3H-a)^2}X_0^2 \tag{11}$$

and

$$T_{max}^b = \frac{M_b\omega^2}{2}x_1^2(z=a) = \frac{M_b\omega^2}{2}X_0^2\frac{a^2}{(3H-a)^2}. \tag{12}$$

The eigenfrequency $\omega_{sb}^2$ is found by equating $T_{max} = V_{max}$ :

$$\omega_{sb}^2 = \frac{420EI(3H-a)}{a^2m_s\left[105H^3 - 105aH^2 + 35Ha^2 - 2a^3 + \frac{140a^2M_b}{m}\right]}. \tag{13}$$



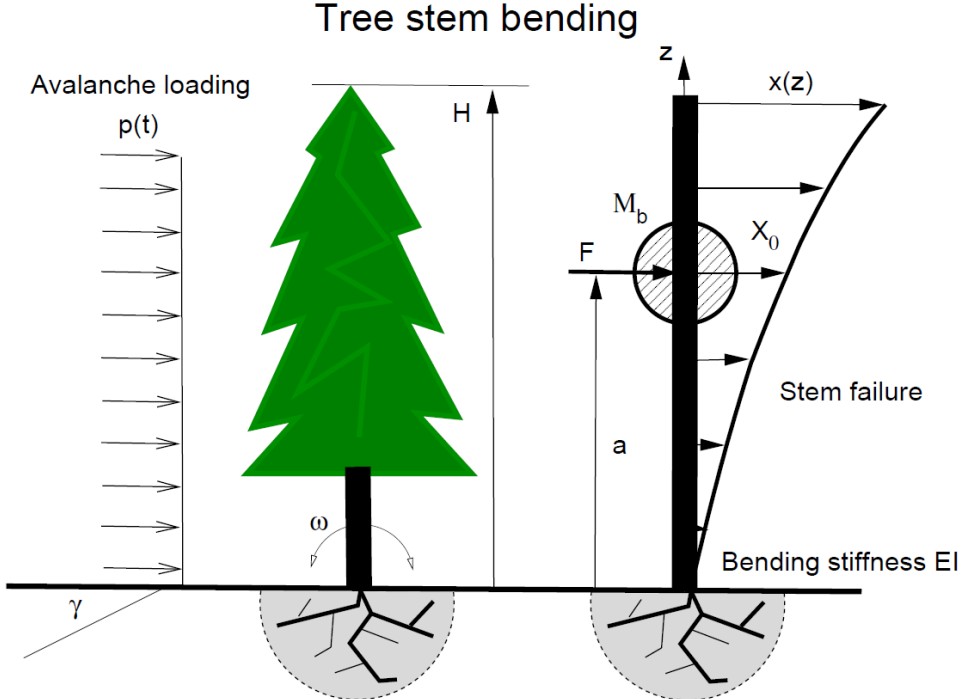

**Figure 2.** A tree of height $H$ breaks in bending. The avalanche exerts a loading $p(t)$ of known (but short) duration. The load acts in the center-of-mass of the tree located a distance $a$ from the ground. The mass of the linear distributed mass of the tree stem is $m_t$ and the lumped mass of the branches is $M_b$. Tree deformation is given by the non-linear distribution $x(z)$.

## 2.2 Eigenfrequency: Tree overturning mode

For the tree overturning case,

$$x(z) = X_0\psi(z) = \frac{FaH}{k}\left[\frac{z}{H}\right].\tag{14}$$

where $k$ is the overturning stiffness of the root-plate. The maximum potential strain energy (overturning) is then

5 $$V_{max} = \frac{1}{2}FX_0 = \frac{1}{2}\frac{k}{aH}X_0^2.\tag{15}$$

Similar to the bending case, the maximum kinetic energy is found by considering the stem and branch energies separately:

$$T_{max}^s = \frac{\omega^2}{2}\int_0^H m_s x_1^2(z)dz = \frac{1}{6}m_s\frac{a^3}{H^2}X_0^2\tag{16}$$

and

$$T_{max}^b = \frac{M_b\omega^2}{2}x^2(z=a) = \frac{M_b\omega^2}{2}X_0^2\frac{a^2}{H^2}.\tag{17}$$





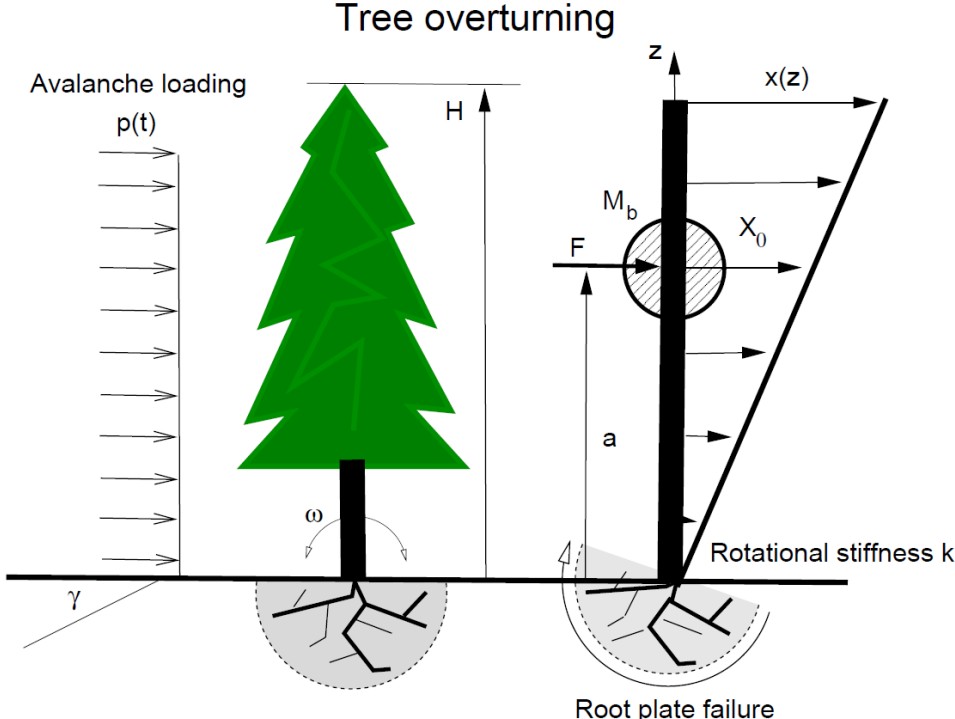

**Figure 3.** A tree of height $H$ breaks by overturning at the root-plate. The avalanche exerts a loading $p(t)$ of known (but short) duration. The load acts in the center-of-mass of the tree located a distance $a$ from the ground. The mass of the linear distributed mass of the tree stem is $m_t$ and the lumped mass of the branches is $M_b$. Tree deformation is given by the linear distribution $v(z)$.

The eigenfrequency $\omega_{ro}^2$ is found by equating $T_{max} = V_{max}$:

$$\omega_{ro}^2 = \frac{3}{[m_s a + 3M_b]} \frac{Hk}{a^3}. \tag{18}$$

## 3   Dynamic magnification of avalanche blast

5   The equation of motion for an undamped system subjected to a harmonic loading is

$$M\ddot{x}(t) + Kx(t) = F(t) = F_0 \sin \overline{\omega} t \tag{19}$$

which has the general solution for $0 \leq t \leq t_0$

$$x(t) = \frac{F_0}{K} \frac{1}{1 - \beta^2} (\sin \overline{t} - \beta \sin \omega t) \tag{20}$$





and for $t > t_0$

$$x(t) = \frac{\dot{x}(t_0)}{\omega} \sin \overline{\omega}(t - t_0) - x(t_0) \sin \omega(t - t_0) \tag{21}$$

where $\beta = \frac{\overline{\omega}}{\omega}$ is the ratio between the frequency of the avalanche blast and eigenfrequency of the tree. The magnitude of the dynamic response therefore depends on the ratio of the load duration to the period of vibration of the tree. For the case when $\beta < 1$ the maximum deformation occurs when the impulsive load is acting. It can be shown, see Clough and Penzien (1975), that the time to this peak response $t_{max}$ is

$$\overline{\omega} t_{max} = \frac{2\pi\beta}{\beta + 1} \tag{22}$$

which can be substituted into the general solution to find the dynamic magnification factor for a long duration impulse:

$$D = \frac{1}{1 - \beta^2} \left[ \sin \overline{\omega} t_{max} - \beta \sin \frac{\overline{\omega} t_{max}}{\beta} \right]. \tag{23}$$

It can be likewise shown that the maximum response for the free vibration case occurs when $\beta > 1$, $t > t_0$. For this case the dynamic magnification factor for a short duration impulse is

$$D = \frac{2\beta}{1 - \beta^2} \cos \frac{\pi}{2\beta}. \tag{24}$$

For the resonance case $\beta = 1$

$$D = \frac{\pi}{2}. \tag{25}$$

## 4 Application

To demonstrate how the dynamic magnification factor $D$ can be found we consider the following problem: a powder snow avalanche enters a spruce forest with considerable speed ($> 50$ m/s) and exerts a short duration air-blast with frequecy $\overline{\omega}$. The duration of the blast is on the order of a few seconds. The height of the trees is between 25m and 30m, which is also the height of powder cloud. The cloud has decoupled from the avalanche core which has stopped before reaching the forest. Moreover, the only loading on the trees is the air-blast.

Using the measured mass values tabulated in Table 1, we set the total branch and needle mass of a single tree to be $M_b = 540$ kg. The stem mass per length is approximately 60 kg/m (wood density 480 kg/m$^3$). The total force of the avalanche impact acts at the tree's center-of-mass which is located $a$=16.5 m above ground. This allows us to define the natural frequency in bending of the tree by Eq. 13, $\omega_{sb} = 1.48$ rad/s (0.24 Hz). This value is in very good agreement with measurements, see Jonsson et al. (2007). The modulus of elasticity was set to $E = 10$ GPa based on experimental measurements (Haines et al., 1996). For the calculations a tree diameter somewhat smaller than the DBH diameter is selected. In this case $d = 0.2$m, which is 1/2 of the DBH diameter (this provides the best match to the experimental frequencies).

Consider first a duration sine impulse lasting 2.50s ($\overline{\omega} = \pi/6$). In this case $\beta = 0.699$; that is, the maximum deformation occurs during the time the load is acting. For this case, application of Eq. 3, we find $D$=1.76, a rather large magnification



**Table 1.** Numerical values for mass distribution of spruce for different tree heights. Table is constructed from data contained in (Indermühle, 1978; Kalberer, 2006; Kramer, 1988). The stated values represent average values for spruce trees in alpine environments. Values are approximate and will change depending on location in forest, slope exposition, etc. Branch mass includes needle mass which is given in parenthesis. Intercepted snow mass is not included in the calculations.

| Height $H$ | Center-of-mass $a$ | Width $w$ | Stem DBH $d$ | Stem mass $m_t$ | Branch mass $M_b$ |
|---|---|---|---|---|---|
| m | m | m | m | kg | kg |
| 3 | 1.80 | 2.0 | 0.10 | 3 | 4 (2) |
| 15 | 8.60 | 3.0 | 0.20 | 20 | 155 (60) |
| 22 | 13.9 | 3.5 | 0.30 | 30 | 310 (120) |
| 27 | 16.3 | 4.5 | 0.40 | 60 | 540 (200) |
| 35 | 21.2 | 7.0 | 0.70 | 150 | 1640 (640) |

**Table 2.** Natural frequencies in bending and overturning for spruce trees of different heights. $E$ = 10 GPa. A reduced stem diameter $d = 0.5 d_{DBH}$ produces a good agreement to measured frequencies. Mass distribution taken from Table 1.

| Height $H$ | Center-of-mass $a$ | $\omega_{sb}$ | $\omega_{ro}$ | $\omega_{ro}$ |
|---|---|---|---|---|
| m | m | rad/s (Hz) | rad/s (Hz) | rad/s (Hz) |
| | | | $k$ = 100 kNm | $k$ = 1000 kNm |
| 3 | 1.80 | 18.20 (2.90) | 104.00 (16.55) | 328.88 (52.34) |
| 15 | 8.60 | 2.09 (0.33) | 3.77 (0.60) | 11.93 (1.90) |
| 22 | 13.9 | 1.45 (0.23) | 0.52 (0.24) | 4.82 (0.76) |
| 27 | 16.3 | 1.48 (0.23) | 0.99 (0.16) | 3.15 (0.05) |
| 35 | 21.0 | 1.65 (0.26) | 0.43 (0.07) | 1.36 (0.21) |

factor. For a shorter duration impulse lasting 1.66s, $\beta$ = 1.27 and from Eq. 3, we find $D$=1.36. The primary conclusion to draw from this analysis is that the natural frequency in bending of tall trees is close to the frequency of the applied avalanche air-blast. Measurements of air-blast duration times reported by Russian researchers are within this range, lasting only a few seconds, see (Grigoryan and others, 1982; Sukhanov and Kholobayev, 1982; Sukhanov, 1982).

5    Measurements of root plate stiffness are rare; however, values for 10m - 14m high spruce reported in Neild and Wood (1998) vary between $k$= 80 kN m ($H$ = 10 m) and $k$ = 1200 kN m ($H$ = 14 m). These values suggest a large variation in $k$ depending on growth conditions. The application of these $k$ stiffness values for spruce trees predicts natural frequencies for root-plate overturning in $\omega_o > 2Hz$ (Eq. 2.2). The calculated $\beta$ factors for overturning are typically $\beta <1$. This result suggests that large dynamic magnification factors can only be generated by very short duration impulses (less than $t < 0.5s$). Tall trees ($H > 20$m) 10    with low root plate stiffnes ($k \approx 100$ kNm) are vulnerable to powder avalanche air-blasts.

## 5   Conclusions

We draw several conclusions from our analysis.

    Firstly, the natural frequency of tall trees – in bending and overturning – is close to the loading frequency of powder avalanches, $\omega \approx \bar{\omega}$. Thus, tall trees are susceptible to powder avalanche blow-down. When using tree-blow down to estimate



avalanche impact pressures (and therefore speed and density of the powder cloud) a dynamic magnification factor should be applied in the analysis. Moreover, powder avalanches can knock down trees with lower velocity than is presently assumed. This result is also valid for other types of tall structures, including power pylons, or buildings with long over-hanging roofs.

Secondly, both tree bending and root-plate overturning are possible tree failure modes when hit by a powder avalanche. Interestingly, the natural frequencies of tree bending and root-plate overturning are similar, when the root-plate stiffness is low ($k < 100$ kN m) and the tree is tall ($H > 20$m). Although there is considerable data available to constrain the value of the modulus of elasticity of wood $E$, there is less information available to constrain the root-plate stiffness. In future, field investigations that document forest destruction should clearly separate bending and overturning failures. This would help understand the variability of tree anchorage on mountain slopes. The field examinations should also quantify the stem diameter $d$ at more than one location as this is necessary to accurately determine the bending eigenfrequency.

Finally, the fact that tall trees can be broken in bending and overturning indicates the nature of the avalanche air blast. It appears to be a high velocity, short duration pulse of flowing material (ice-dust), similar to a high-density gust of wind. It is not a compression wave travelling at the speed of sound.

*Acknowledgements.* This work was performed within the framework of the joint Austrian-Swiss project bDFA, a study of avalanche motion beyond the dense flow avalanche regime. We thank the Austrian Academy of Science (ÖAW) for their financial support as well as the Austrian research partners (Austrain Research Centre for Forests, Torrent and Avalanche Control and the Universtiy of Innsbruck).



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
