# Peer review of "Brief Communication: Dynamic magnification factors for tree blow-down by powder snow avalanche air blasts"

_Natural Hazards and Earth System Sciences, 2017_

## Referee Comment (RC1) · Anonymous Referee #1 · 28 Sep 2017

General comments: The manuscript describes a simple method to determine the dynamic response of tress to impulsive loads, in particular related to avalanche impact. Although there exist a few papers dealing with forest damage due to snow avalanches, less attention was drawn to the powder avalanche air blasts. But those effects are of great relevance for quantifying powder avalanche pressures from case studies. Main objective of this manuscript was to identify how short duration powder avalanche blasts can break and overturn trees. The authors found that tall trees are prone to avalanche air blasts; dynamic magnification factors should be considered when back-calculating avalanche impact pressures. Although further field examinations will be necessary, this paper is a very good first approach; it can be accepted with minor revisions.

[Figure]

Specific comments and technical corrections:

page 1, line 22: it should mean: '. . .figs. 2 und 3.'

page 2, line 1: it should mean: 'The eigenfrequency of the tree is a function of. . .'

page 3, line 29: it should mean: '. . . from the ground (see figures 2 and 3).'

page 4, line 17: it should mean: '. . . is composed of two parts containing the stem energy Tsmax an the branch energy Tbmax of the tree, ...'

page 4, equation 12: I assume it should be '$\omega sb2$' instead of '$\omega 2$'

page 4, equation 13, last term in the denominator: it should mean 'ms' instead of 'm'

page 5, equation 17: I assume it should be '$\omega ro2$' instead of '$\omega 2$'

page 7, line 29: it should mean: '. . . application of Eq. 23, we find. . .'

page 8, Table 1: in the fifth column it should mean 'kg/m' instead of 'kg'

page 8, line 1: it should mean: '. . . and from Eq. 23, we find. . .'

page 8, line 14: it should mean: 'When using tree blow-down to estimate. . .'

---

## Referee Comment (RC2) · Anonymous Referee #2 · 28 Nov 2017

Dear editor,

The paper presents an elegant method to estimate the dynamic magnification factor for a tree blow-down by a powder avalanche. Considering this factor, is important when back-calculating the avalanche parameters that causes the tree failure. Ignoring this factor may increase the estimated velocities and densities of powder snow avalanches and therefore making error in assessing the destruction power of this natural hazard. By making some basic assumptions and solving the equations analytically the authors improve our quantitative analysis of powder avalanches. In general, the paper is well written and organized. My main comments are to add some references that support

the use of basic models and to explain a bit more some of the terms and ideas. I hope to see it published after the authors will address these minor comments. Hope this helps

Minor comments:

Page 1, lines 11-12: "The age of the destroyed trees can be additionally used to link the historical observations to avalanche return period." Is there any previous study you can cite here?

Page 1, line 22: "We assume two deformation modes: stem bending and root-plate overturning" As these two modes of failure are basic terms in this paper, I would describe in more details what is the different in the failure mechanism between the two. Can we say that failure by bending causes by stem breakage at the base of the tree? Is it the same style of failure in the case of root-plate overturning or it is just tearing of many small roots? What are the field observations that support each deformation mode?

Page 1, line 24: Regarding the dynamic magnification factor. "This value is used to magnify the non-impulsive loadings". This is the most important term in the paper and you define it before. However, is it possible to show shortly, but somewhat quantitatively, how this factor actually influences the probability for failure. In the paper we get the method how to calculate D but what is the next step? Can the authors suggest a short answer for it in the text?

Page 1, line 24: A typo: change "he" to "the".

Page 3, line 6: I think that factor 2 is missing in the equation. Should be: $\omega = 2\pi f$

Page 3, line 8: Are you the first to use drag force on trees? If not please cite a previous study. It is a turbulent drag equation as the powder avalanche is a fluid turbulence and therefore the drag equation is the one in use. I would emphasize this point and add some basic citation.

[Figure]

Page 4, equations 6 & 7: Did you define $\psi$?

Page 4, lines 6-7: What is the equation that is solved to get equations 6 & 7? Please add a reference here!

Page 4, line 13: "The maximum potential strain energy in bending is". Please add a reference here.

Page 4, equation 11: Same for here, what is the definition for the kinetic energy (shouldn't it includes a velocity term?). Please add a reference.

Page 5, line 4: I would add a short explanation of the conceptual model as was explained for the bending case (just below equations 6 & 7).

Page 7, line 26: What is DBH? Please define it in text.

[Figure]

---

## Author Comment (AC2) · 3 Jan 2018

**RESPONSE TO REVIEWER 1**

General comments: The manuscript describes a simple method to determine the dynamic response of tress to impulsive loads, in particular related to avalanche impact. Although there exist a few papers dealing with forest damage due to snow avalanches, less attention was drawn to the powder avalanche air blasts. But those effects are of great relevance for quantifying powder avalanche pressures from case studies. Main objective of this manuscript was to identify how short duration powder avalanche blasts can break and overturn trees. The authors found that tall trees are prone to avalanche air blasts; dynamic magnification factors should be considered when back-calculating avalanche impact pressures. Although further field examinations will be necessary, this paper is a very good first approach; it can be accepted with minor revisions. Specific comments and technical corrections:

**RESPONSE:** We sincerely thank the reviewer: the reviewer found some mistakes in the notation, especially in the equations, and we truly appreciate the careful reading. We are very thankful that the reviewer grasped the idea behind the paper. The problem of using tree destruction to back-calculate avalanches is a very common problem for us. The idea of dynamic impact factors is not well known in the avalanche community.

page 1, line 22: it should mean: '. . .figs. 2 und 3.' → **CHANGED**
page 2, line 1: it should mean: 'The eigenfrequency of the tree is a function of. . .'→ **CHANGED**
page 3, line 29: it should mean: '. . . from the ground (see figures 2 and 3).'→ **CHANGED**
page 4, line 17: it should mean: '. . . is composed of two parts containing the stem energy Tsmax an the branch energy Tbmax of the tree, ...' → **CHANGED**
page 4, equation 12: I assume it should be '!sb2' instead of '!2'→ **Yes, CHANGED. Also changed Eq. 11**
page 4, equation 13, last term in the denominator: it should mean 'ms' instead of 'm' → **YES, CHANGED!**
page 5, equation 17: I assume it should be '!ro2' instead of '!2'→ **CHANGED, also changed Eq. 16.**
page 7, line 29: it should mean: '. . . application of Eq. 23, we find. . .' → **YES, CHANGED**
page 8, Table 1: in the fifth column it should mean 'kg/m' instead of 'kg' → **YES; CHANGED**
page 8, line 1: it should mean: '. . . and from Eq. 23, we find. . .' → **Yes, it should be Eq. 24**
page 8, line 14: it should mean: 'When using tree blow-down to estimate. . . → **CHANGED.**

**RESPONSE TO REVIEWER 2**

The paper presents an elegant method to estimate the dynamic magnification factor for a tree blow-down by a powder avalanche. Considering this factor, is important when back-calculating the avalanche parameters that causes the tree failure. Ignoring this factor may increase the estimated velocities and densities of powder snow avalanches and therefore making error in assessing the destruction power of this natural hazard. By making some basic assumptions and solving the equations analytically the authors improve our quantitative analysis of powder avalanches. In general, the paper is well written and organized. My main comments are to add some references that support the use of basic models and to explain a bit more some of the terms and ideas. I hope to see it published after the authors will address these minor comments. Hope this helps.

RESPONSE: We thank the reviewer for his constructive suggestions and have made the following changes:

Minor comments:
Page 1, lines 11-12: "The age of the destroyed trees can be additionally used to link the historical observations to avalanche return period." Is there any previous study you can cite here? → **Good point. We cite the following papers, which we have added to references**

**Schläppy, R., Eckert, N., Jomelli, C., Stoffel, M., Grancher, D. Brunstein D., Naaim, M., Deschatres, M. 2014. Validation of extreme snow avalanches and related return periods derived from a statistical-dynamical model using tree-ring techniques. Cold regions and technology 99: 12-26.**

Reardon, B. A., G. T. Pederson, Caruso, C.J, Fagre, D.B. 2008. Spatial Reconstructions and Comparisons of Historic Snow Avalanche Frequency and Extent Using Tree Rings in Glacier National Park, Montana, U.S.A." Arctic, Antarctic and Alpine Research 40: 148-160.

Gądek, B., Kaczka, R. J., Rączkowska, Z., Rojan, E., Casteller, A., and Bebi, P.: Snow avalanche activity in Żleb Żandarmerii in a time of climate change (Tatra Mts., Poland), CATENA, 158, 201–212, 2017.

Page 1, line 22: "We assume two deformation modes: stem bending and root-plate overturning" As these two modes of failure are basic terms in this paper, I would describe in more details what is the different in the failure mechanism between the two. Can we say that failure by bending causes by stem breakage at the base of the tree? Is it the same style of failure in the case of root-plate overturning or it is just tearing of many small roots? What are the field observations that support each deformation mode? → **CHANGED**
 **We introduced the following text at around line 17 (before line 22) with extensive citations: "Tree-breaking depends on both the avalanche loading and tree strength. Trees fall if the bending stress exerted by the avalanche exceeds the bending strength of the tree stem (Johnson, 1987; Mattheck and Breloer, 1994, Peltola et al., 1997, 1999;) or if the applied torque overcomes the strength of the root-soil plate, leading to uprooting and overturning (Coutts, 1983; Jonsson et al., 2006). Both mechanisms depend on the local flow height of the avalanche."**

Coutts, M.: Root architecture and tree stability, Plant Soil, 71, 171–188, 1983.

Mattheck, C. and Breloer, H.: Handbuch der Schadenskunde von Bäumen: Der Baumbruch in Mechanik und Rechtsprechung, Freiburg im Breisgau: Rombach, 1994

Peltola, H., Kellomäki, S., Väisänen, H., and Ikonen, V. P.: A mechanisticmodel for assessing the risk of wind and snow damage to single trees and stands of scots pine, norway spruce, and birch,Can. J. Forest Res., 29, 647–661, 1999.

Johnson, E. A.: The relative importance of snow avalanche disturbanceand thinning on canopy plant populations, Ecology, 68,43–53, 1987.

Jonsson, M., Foetzki, A., Kalberer, M., Lundström, T., Ammann, W., and Stöckli, V.: Root-soil rotation stiffness of norway spruce (Picea Abies (l.) karst) growing on subalpine forested slopes, Plant Soil, 285, 267–277, 2006.

Peltola, H., Nykänen, M. L., and Kellomäki, S.: Model computations on the critical combination of snow loading and windspeed for snow damage of scots pine, norway spruce and birch sp. At stand edge, Forest Ecol. Manag., 95, 229–241, 1997.

Page 1, line 24: Regarding the dynamic magnification factor. "This value is used to magnify the non-impulsive loadings". This is the most important term in the paper and you define it before. However, is it possible to show shortly, but somewhat quantitatively, how this factor actually influences the probability for failure. In the paper we get the method how to calculate D but what is the next step? Can the authors suggest a short answer for it in the text? → **MODIFIED:  We have added the following text to try to help the reader quantify this effect. "As we shall see, an error of up to 25% can be made when back-calculating avalanche velocities.  For example, an avalanche travelling at 35 m/s exerts the same pressure as an avalanche travelling at 50 m/s if the impulsive nature of the loading is considered. These are significant differences in hazard mitigation studies.**
Page 1, line 24: A typo: change "he" to "the". → **CHANGED**
Page 3, line 6: I think that factor 2 is missing in the equation. Should be: $!=2\_f$: →**NO CHANGE: t0 is HALF the PERIOD, so there is no factor 2!  Good check.**
Page 3, line 8: Are you the first to use drag force on trees? If not please cite a previous study. It is a turbulent drag equation as the powder avalanche is a fluid turbulence and therefore the drag equation is the one in use. I would emphasize this point and add some basic citation. → **Good point.  I cite Bozhinskiy, and Losev which has an entire chapter dedicated to this problem. Also the recent work of Feistl.**
Page 4, equations 6 & 7: Did you define  psi. → **psi is the interpolation function.  It is now defined. We write "The functions $\psi_1(z)$ and $\psi_2(z)$ represent interpolation functions for the deformation field."**

Page 4, lines 6-7: What is the equation that is solved to get equations 6 & 7? Please add a reference here! **→ These equations can be found in Clough and Penzien and therefore we add this reference.**

Page 4, line 13: "The maximum potential strain energy in bending is". Please add a reference here. **→ The book by Clough and Penzein has an entire chapter on Rayleigh analysis. Added reference.**

Page 4, equation 11: Same for here, what is the definition for the kinetic energy (shouldn't it includes a velocity term?). Please add a reference. **→ The book by Clough and Penzein has an entire chapter on Rayleigh analysis. Added reference. The equation DOES contain a velocity in the form of omega^2 (the vibration velocity).**

Page 5, line 4: I would add a short explanation of the conceptual model as was explained for the bending case (just below equations 6 & 7). **→ We have added a conceptual description (torsional spring stiffness) and a new citation.**

Page 7, line 26: What is DBH? Please define it in text. **→ It stands for stem "Diameter at Breast Height". It is well known to tree experts. We have added definition to text**